

# iSeqSearch: incremental protein search for iBlast/iMMSeqs2/iDiamond

Hyunwoo Yoo[1], Mohammadsaleh Refahi[1], Robi Polikar[2], Bahrad A. Sokhansanj[1], James R. Brown[1] and Gail L. Rosen[1]

[1] Department of Electrical and Computer Engineering, Drexel University, Philadelphia, PA, United States of America
[2] Electrical and Computer Engineering, Rowan University, Glassboro, NJ, United States of America

## ABSTRACT

**Background**. The advancement of sequencing technology has led to a rapid increase in the amount of DNA and protein sequence data; consequently, the size of genomic and proteomic databases is constantly growing. As a result, database searches need to be continually updated to account for the new data being added. However, continually re-searching the entire existing dataset wastes resources. Incremental database search can address this problem.

**Methods**. One recently introduced incremental search method is iBlast, which wraps the BLAST sequence search method with an algorithm to reuse previously processed data and thereby increase search efficiency. The iBlast wrapper, however, must be generalized to support better performing DNA/protein sequence search methods that have been developed, namely MMseqs2 and Diamond. To address this need, we propose iSeqsSearch, which extends iBlast by incorporating support for MMseqs2 (iMMseqs2) and Diamond (iDiamond), thereby providing a more generalized and broadly effective incremental search framework. Moreover, the previously published iBlast wrapper has to be revised to be more robust and usable by the general community.

**Results**. iMMseqs2 and iDiamond, which apply the incremental approach, perform nearly identical to MMseqs2 and Diamond. Notably, when comparing ranking comparison methods such as the Pearson correlation, we observe a high concordance of over 0.9, indicating similar results. Moreover, in some cases, our incremental approach, iSeqsSearch, which extends the iBlast merge function to iMMseqs2 and iDiamond, provides more hits compared to the conventional MMseqs2 and Diamond methods.

**Conclusion**. The incremental approach using iMMseqs2 and iDiamond demonstrates efficiency in terms of reusing previously processed data while maintaining high accuracy and concordance in search results. This method can reduce resource waste in continually growing genomic and proteomic database searches. The sample codes and data are available at GitHub and Zenodo (https://github.com/EESI/Incremental-Protein-Search; DOI: 10.5281/zenodo.14675319).

Corresponding author
Gail L. Rosen, glr26@drexel.edu

## INTRODUCTION

In the realm of genomic and proteomic research, the advent of high-throughput sequencing technologies has precipitated an unprecedented expansion in the volume of nucleic acid and amino acid sequence data. This deluge of data has, in turn, necessitated the development and expansion of comprehensive sequence databases to catalog and make sense of the wealth of information. However, the traditional approach of exhaustive search within ever-growing sequence data repositories poses a significant challenge in terms of computational efficiency and resource allocation.

This rapid accumulation of sequence data is not merely a technical challenge but a fundamental shift in our ability to understand biological systems. The exponential growth of databases like UniProtKB/Swiss-Prot, which now contain millions of protein sequences, offers new opportunities for discovering evolutionary relationships, predicting protein structures, and understanding molecular functions. However, the growth of databases also presents substantial computational hurdles that demand innovative solutions.

We recently introduced Naïve Bayes taxonomic classifier (NBC++), an incremental taxonomic Naïve Bayes classifier that updates its database with new data (*Zhao, Cristian & Rosen, 2020*). The novelty of NBC++ is that the classifier does not need to be completely retrained when the training data is expanded. However, in the original NBC++, queries previously classified using the database results are not automatically updated with this incremental approach. Subsequently, an incremental search method using BLAST, known as iBlast by *Dash et al. (2021)*, was developed. This method automatically updates hit results by merging the previous hits with the results retrieved when searched against new data. Therefore, iBlast reduces the computational burden by leveraging previously processed data within the BLAST framework (*Altschul et al., 1990*). Despite its utility, the advent of more advanced and efficient search tools such as Diamond (*Buchfink, Reuter & Drost, 2021*) and MMseqs2 (*Steinegger & Söding, 2017*), which offer superior performance in handling large-scale sequence data, has rendered the traditional BLAST-based methods less optimal. BLAST is generally slower than modern tools, such as MMSeqs2 and Diamond, and may identify fewer hits under certain search conditions, particularly for large and complex datasets. An important observation is that none of these modern tools, including BLAST, MMseqs2, and Diamond, natively support incremental search. For BLAST, incremental functionality is only available through a standalone module called iBlast, which is restricted to handling XML outputs. Similarly, MMseqs2 provides incremental capabilities exclusively for clustering tasks, highlighting a key limitation: while these tools excel in speed and scalability, they lack built-in support for incremental search.

Here we introduce a methodology that integrates the incremental search principle of iBlast with the advanced search capabilities of Diamond and MMseqs2. (Note that while MMSeqs2 has an incremental clustering ability (*Steinegger & Söding, 2017*; *Nguyen et al., 2023*), we focus on incremental search). Our approach is designed to enhance search efficiency in the face of continuously expanding sequence databases, thereby ensuring sustainable and effective data management and retrieval. Specifically, we present a novel approach that allows for rapid integration of new sequences into existing search results,

without the need for complete database re-scans. Furthermore, we introduce an extended file format, m8e, which enhances the standard m8 format by incorporating critical metadata.

## MATERIALS & METHODS

We propose an incremental method, called iSeqSearch, for efficient protein sequence similarity searches, utilizing the Scope Astral protein database provided by *Fox, Brenner & Chandonia (2014)* as our benchmark dataset. This study implements an incremental method that utilizes Spouge statistics by *Park et al. (2012)*. Our approach is similar to the iBlast method which maintains accuracy while improving the processing speed for new data. The Spouge statistics method used in this study is directly adopted from the iBlast framework. The novelty of iSeqSearch lies in applying iBlast's incremental approach, including its use of Spouge statistics, to MMseqs2 and Diamond. By integrating Spouge statistics with these widely used modern tools, we enable the benefits of iBlast's incremental process in new contexts. Our method employs a two-step process: (1) initial database search using BLAST, MMseqs2, or Diamond with their default parameters, and (2) incremental updates using our custom algorithm based on Spouge statistics. Portions of this text were previously published as part of a preprint (https://www.biorxiv.org/content/10.1101/2024.09.09.612094v1).

Instead of complex XML files, iSeqSearch uses the m8 or m8e file format for data merging, simplifying the conversion and integration of these files. The m8 format contains tab-separated fields: query id, subject id, percent identity, alignment length, mismatches, gap opens, q. start, q. end, s. start, s. end, evalue, and bit score. Additionally, we propose an extended version of the m8 file format, herein named m8e (m8 extension). The m8e format includes the total length of the database within the file, adding an extra line at the beginning of each file to store this information. We note that the m8e format is identical to the m8 file format, except for the first line. The first line explicitly specifies the length of the database, eliminating the need to recalculate the database length during the merging function. This simple addition offers an outsized advantage of improved processing speed. The database length is crucial for accurate *E*-value calculations during incremental updates, and the m8e format allows for the automatic integration of two result files. The m8e format also eliminates the need for separate measurements of DB length, saving time and improving overall efficiency. The Supplementary Material provides examples to illustrate the new format.

We implemented our method in Python (version 3.8+), utilizing the Biopython library (version 1.79) for sequence handling and NumPy (version 1.21.0) for numerical operations. To evaluate the incremental method's performance, we conducted comprehensive benchmarks comparing our incremental approach against full database searches. To do so, we divided the Scope Astral protein database into 10 batches, with the first batch stratified based on protein class and the remaining nine batches all stratified in the same way as the first batch (and then the individual batches randomly partitioned). The first batch serves as the query set, while the remaining nine batches were sequentially
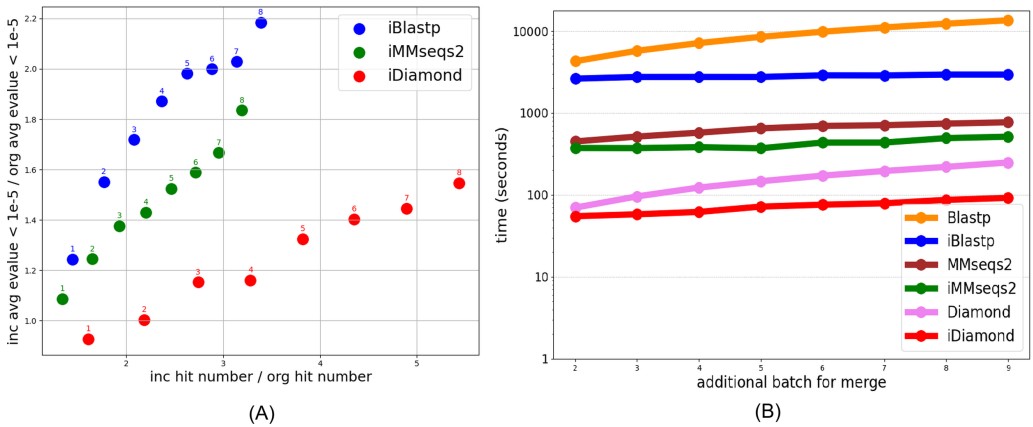

**Figure 1** The left graph (A) illustrates increasing e-values and hit counts when comparing incremental methods (iSeqsSearch including iBlastp, iMMseqs2, iDiamond) to non-incremental methods (Blastp, MMseqs2, Diamond). The x-axis represents the ratio of the number of hits identified by incremental methods to those identified by non-incremental methods. The y-axis represents the ratio of the average e-values obtained by incremental methods to those obtained by non-incremental methods. In this graph, only e-values below 1e-5 are considered. The right graph (B) compares the processing times of these methods, showing that incremental methods are faster. Additional Batch for Merge refers to a scenario in incremental search where the database size increases progressively with the addition of multiple batches. The value of Additional Batch for Merge determines the number of batches contributing to the database size.

combined to form increasingly larger search databases, simulating database growth over time. Experiments used Blastp, MMseqs2, and DIAMOND with default settings unless otherwise stated (changing the hits returned under an e-value threshold and also extending the limit of the returned hits beyond the default limits), only adjusting the number of threads to 32 for consistency. For each database size (from 2 to 10 batches), we performed both full and incremental searches.

## RESULTS

Our analysis of the incremental method for protein sequence searches revealed improvements in both search effectiveness and computational efficiency compared to traditional non-incremental methods.

Figure 1A illustrates two primary findings: First, iSeqsSearch methods including iBlastp, iMMseqs2, and iDiamond consistently yield higher hit counts compared to their non-incremental counterparts (Blastp, MMseqs2, Diamond). This increase in hits suggests that incremental methods are capable of identifying additional potential matches, thereby enhancing the comprehensiveness of the search. The e-values of the additional hits fall within acceptable ranges, ensuring that these matches are statistically significant and biologically plausible. Second, the processing times for incremental methods are substantially reduced, as shown in the right graph of Fig. 1B. This time efficiency is crucial for large-scale protein database searches where computational resources are often a limiting factor.

It is important to note that while incremental methods result in increased hit counts, they also lead to larger e-values (Fig. 1A). Although larger e-values typically indicate less statistically significant matches, we only include hits below the commonly used threshold of 1e-5 in Fig. 1A). According to *Choudhuri (2014)*, e-values less than and equal to 1e-5 are typical thresholds for judging homology, suggesting that additional matches, while potentially less statistically confident, may still be biologically relevant and worthy of consideration in many applications. As a case study, we examine one type of protein for hits that have the "correct"/"right" protein family label *vs.* hits that have incorrect labels. This case study demonstrates the utility of incremental methods in identifying biologically meaningful hits across diverse protein folds, highlighting their adaptability to larger datasets. Further details of the case study, including the dataset and analysis methodology, are provided in the Supplementary Material. In general, the incremental version had higher e-values than the non-incremental version. For instance, while Blast and MMSeqs2 (Figs. S22 and S23) returned more hits per batch, the hits returned by Diamond (Fig. S24) were fewer but had much lower e-values, indicating greater statistical significance.

When incremental batch learning is used, there is a reduction in queries with no hits. This reduction further underscores the effectiveness of the incremental method in improving search coverage. As shown in Fig. 15 in the Supplementary Materials, the proportion of queries with zero hits using incremental methods is lower than that of non-incremental methods, illustrating the potential capability of incremental methods to cover a broader range of relevant sequences.

To assess the quality and consistency of search results, we show the results of several correlation measures. While there are several similarity measures available, including the Pearson correlation coefficient (*Pearson, 1896*), the Kendall tau correlation coefficient (*Kendall, 1938*), and the Spearman correlation coefficient (*Spearman, 1904*), here we utilize: (1) Pearson to assess the concordance of the search methods in their e-value and (2) Kendall tau to assess the concordance between the methods' rankings. Figure 2 presents heatmaps of the Kendall tau and Pearson correlation coefficients for each search method. The Pearson correlation heatmap reveals high similarity between incremental and non-incremental methods in the e-values of overlapping hits between methods, with iBlastp and Blastp achieving a score of 0.97, while iMMseqs2 and MMseqs2, as well as iDiamond and Diamond, show perfect correlation with a score of 1.0 (Fig. 2). These high correlation scores indicate that incremental methods maintain a quality of results comparable to their non-incremental counterparts.

The Kendall tau correlation, which measures rank consistency, also shows strong agreement between incremental and non-incremental methods. iBlastp and Blastp score 0.57, while iDiamond and Diamond achieve 0.48, both higher than other method pairs (Fig. 2). These results suggest that the incremental methods preserve the ranking of hits, an important factor in sequence similarity searches.

The quality of the searches is also measured using log discounted cumulative gain by *Järvelin & Kekäläinen (2002)* and normalized discounted cumulative gain by *Wang et al. (2013)*. We evaluate the search quality using log discounted cumulative gain (DCG). The consistent increase in log DCG values across incremental methods indicates

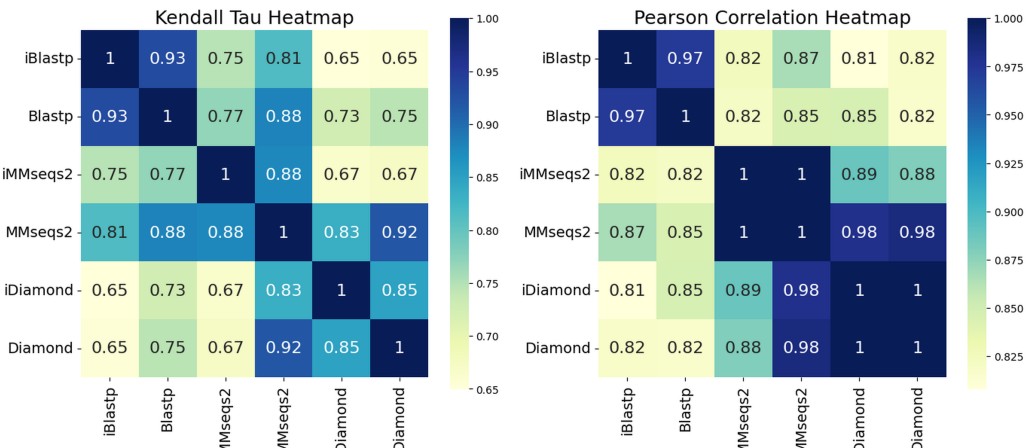

**Figure 2** **Heatmaps representing the Kendall tau correlation coefficient and the Pearson correlation coefficient for each search result.** In the searches, 1/10th of the Scope Astral protein dataset is used as the query, and the remaining dataset is used as the database. The database is randomly sampled and divided into nine equal batches, which are then incrementally combined based on the search results of the fully combined dataset. In the Pearson correlation heatmap, iBlastp and Blastp achieve a score of 0.97, which was the highest among the methods, while iMMseqs2 and MMseqs2, as well as diamond and Diamond, show a score of 1.0, indicating the highest similarity. For the Kendall tau correlation, iBlastp and Blastp scored 0.93, which was higher than the other methods, and iDiamond and Diamond also had a higher score of 0.85. These observations indicate that iBlastp and Blastp, iMMseqs2 and MMseqs2, and iDiamond and Diamond all provide similar results.

improvements in the quality of search results, reflecting both better ranking of hits and more effective identification of relevant protein sequences as shown in Fig. S16.

Furthermore, we compare the average number of hits per query for each method with and without maximum target hit limits (Table 1). iBlastp achieves a remarkably high average of 725.93 hits per query, surpassing both Blastp with no limit (253.58 hits) and with the default limit (240.89 hits). Without a maximum limit, Diamond achieves an average of 435.78 hits per query, higher than the 22.14 hits with the default limit and 120.38 hits with iDiamond (which uses default Diamond parameters).

The efficiencies of incremental methods are further highlighted by their reduced search times. For instance, iDiamond completes searches approximately 19 times faster than Diamond without a hit limit. iMMseqs2 was about two times faster, and iBlastp was around 16 times faster than their non-incremental counterparts. Venn diagram analysis (Figs. 12, 13 and 14 in Supplementary Materials) reveals that over 90% of the hits from non-incremental methods are included in the hits from incremental methods, demonstrating that incremental methods maintain comprehensive coverage while improving efficiency.

As a further test of our methodologies, we attempted reclassification of protein queries across seven major classes in the Scope Astral protein database (*Chandonia et al., 2022*). The graphs (Figs. 9, 10, and 11 in Supplementary Material) compare the protein class F1 score of non-incremental methods and incremental methods based on the e-value top hit criterion. The results of the incremental experiments show a consistent increase in

**Table 1** Comparison of average hits (e-value is less than 1e−5) per query and time (seconds) for different methods. Default limits are 25, 300, 500 for Diamond, MMseqs2, Blastp.

| Method | Average hits per query | Time (seconds) | Average hits per time |
|---|---|---|---|
| Blastp (No Limit) | 253.58 | 48000 | 158.90 |
| Blastp (Default) | 240.89 | 13566 | 543.93 |
| iBlastp (each batch avg.) | 446.05 | 2820 | 4831.2198 |
| iBlastp (after all batches run) | 725.93 | 22566 | 983.9676 |
| MMseqs2 (No Limit) | 682.29 | 48160 | 430.86 |
| MMseqs2 (Default) | 166.74 | 771 | 6609.69 |
| iMMseqs2 (each batch avg.) | 316.77 | 435 | 22252.33 |
| iMMseqs2 (after all batches run) | 517.08 | 3381 | 4672.31 |
| Diamond (No Limit) | 435.78 | 1792 | 7428.64 |
| Diamond (Default) | 22.14 | 249 | 2716.60 |
| iDiamond (each batch avg.) | 73.29 | 73 | 30649.67 |
| iDiamond (after all batches run) | 120.38 | 581 | 6331.61 |

F1 score across all cases, with classification performance improving steadily as more data is added. The performace reaches 99% when data for all Scope Astral protein classes is included. We also observed a similar trend of increasing F1 scores for both incremental and non-incremental methods. While the performance of incremental methods was sometimes slightly better (*e.g.*, iMMseqs2) or slightly worse (*e.g.*, iDiamond), the overall trend remains similar.

An additional experiment was conducted that queries an experimentally-obtained metatranscriptome obtained from wastewater samples against a chronologically increasing Swiss-Prot database from the years 2005, 2010, 2015, 2020, and 2024. For the query, one of the wastewater samples from Southern California by *Rothman et al. (2023)* was selected. The search results of the proposed incremental method, iMMseqs2, were compared against the conventional MMseqs2. The comparison revealed that the results were similar, and consistent with previous experiments: iMMseqs2 demonstrated time savings while providing a greater variety of hit results. Detailed experimental procedures and results are provided in the supplementary material.

## DISCUSSION

Our study confirmed that incremental methods in protein sequence searches provide broader coverage by generating more hits than non-incremental methods. Additionally, by using DCG, a ranking metric often used in recommendation systems, we verified that the additional hits identified by incremental methods are ranked effectively. This observation suggests that incremental methods not only increase the number of hits but also yield qualitatively significant results.

The reduced processing times of incremental methods offer additional substantial advantages, particularly when searching large-scale databases. A potential future area of investigation would be to determine if there are potential performance differences across databases of varying sizes and designs.

While there is a tendency for e-values to increase, we found that applying a 1e-5 threshold provided effective filtering. These results demonstrate that incremental methods, while yielding more hits, offer results with comparable reliability to those of non-incremental methods. Therefore, incremental methods can maintain statistically significant results while encompassing a broader range of hits.

In terms of Scope Astral protein classification, incremental methods showed improved performance in some categories. Notably, iMMseqs2 achieved higher F1 scores, suggesting that incremental methods can also be useful for protein classification tasks. Future research should explore the applicability of incremental methods in other classification systems, such as CATH and Pfam.

Incremental search methods also show potential in addressing the challenges associated with complex, little-studied environmental samples. For instance, these methods can be applied to the analysis of wastewater samples, enabling the detection of a broader range of taxonomic groups that might be missed by conventional approaches. This capability provides valuable insights into microbial diversity and contributes to the identification of potential environmental and public health risks.

## CONCLUSIONS

Our results demonstrate that the incremental method implemented in iSeqSearch offers advantages over traditional sequence search methods. Incremental-Protein-Search using iSeqSearch increases hit counts, reduces computational time, and maintains result quality and ranking consistency. The trade-off between slightly higher e-values and increased hit counts should be carefully considered in the context of specific research goals. Higher e-values can lead to the inclusion of more false positives, which may lower the overall accuracy of the results. The incremental learning method applied to any base search enhances both efficiency and accuracy in large-scale protein database searches, contributes not only to scaling Blast but also other search methods like MMSeqs2 and Diamond. Our proposed approach supports any search tool that outputs results in the m8 file format, making it adaptable to new methods that also utilize this format. In conclusion, iSeqSearch offers the advantage of providing faster and more comprehensive retrieval by leveraging previous results, especially as the size of large-scale databases continues to grow.

## ACKNOWLEDGEMENTS

We utilized OpenAI's ChatGPT4o to assist in drafting and editing parts of this manuscript, particularly in refining the language.

### Funding
This work was supported by the National Science Foundation (No. 1936791, No. 1919691, No. 2107108 and No. 1936782). The funders had no role in study design, data collection and analysis, decision to publish, or preparation of the manuscript.

### Grant Disclosures
The following grant information was disclosed by the authors:
National Science Foundation: 1936791, 1919691, 2107108, 1936782.

### Competing Interests
The authors declare there are no competing interests.

### Author Contributions
- Hyunwoo Yoo performed the experiments, analyzed the data, prepared figures and/or tables, authored or reviewed drafts of the article, and approved the final draft.
- Mohammadsaleh Refahi analyzed the data, prepared figures and/or tables, authored or reviewed drafts of the article, and approved the final draft.
- Robi Polikar conceived and designed the experiments, authored or reviewed drafts of the article, and approved the final draft.
- Bahrad A. Sokhansanj conceived and designed the experiments, authored or reviewed drafts of the article, and approved the final draft.
- James R. Brown conceived and designed the experiments, authored or reviewed drafts of the article, and approved the final draft.
- Gail L. Rosen conceived and designed the experiments, analyzed the data, authored or reviewed drafts of the article, and approved the final draft.

### DNA Deposition
The following information was supplied regarding the deposition of DNA sequences:
The sequences are available at SCOPe under the Astral Sequences & Subsets: Available at https://scop.berkeley.edu/downloads/scopeseq-2.08/astral-scopedom-seqres-gd-all-2.08-stable.fa.

### Data Availability
The Incremental-Protein-Search is available at GitHub and Zenodo:
- Available at https://github.com/EESI/Incremental-Protein-Search
- ece303, & yoohw2. (2025). EESI/Incremental-Protein-Search: iSeqSearch (v1.0.1). Zenodo. Available at https://doi.org/10.5281/zenodo.14675319.

### Supplemental Information
Supplemental information for this article can be found online at http://dx.doi.org/10.7717/peerj.19171#supplemental-information.

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
