# Peer review of "iSeqSearch: incremental protein search for iBlast/iMMSeqs2/iDiamond"

_PeerJ, doi:10.7717/peerj.19171_

## Round 0.1 · original submission · Major Revisions

The reviewers have completed their assessment of the manuscript. Please revise it based on the suggestions and comments. A point-by-point response letter is needed when resubmitting your manuscript for further consideration.

·

Basic reporting

The article is well-structured and provides a clear introduction to the problem of incremental protein searches. The background information outlines the challenges posed by the growing size of genomic and proteomic databases, and the need for incremental search methods. The text is coherent, the methodology is well-documented, and the results are presented in a comprehensible manner.

The figures (e.g., heatmaps and graphs) are visually clear, well-labeled, and enhance the understanding of the results. Figure 1 compares hit counts and processing times between incremental and non-incremental methods, while Figure 2 shows the correlation between various search methods. These figures support the claim that incremental methods such as iBlast, iMMseqs2, and iDiamond perform efficiently in comparison to their non-incremental counterparts.

While the article is well-structured and provides relevant background, it lacks sufficient detail in explaining what sets this work apart from existing incremental search methods. Incremental searches are already a feature in tools like BLAST, MMseqs2, and Diamond, and the authors do not clearly outline how their method differs from or improves upon these built-in features. To better meet reporting standards, the article would benefit from a more thorough explanation of the unique contributions of this work.

If the main innovation lies in the integration of different tools under the iBlast wrapper, this should be explicitly highlighted, along with a clear discussion of the practical advantages this integration offers. Additionally, the introduction of the new m8e format is mentioned but not sufficiently detailed in terms of its impact and how it improves over existing formats. Providing these clarifications would help to better frame the article’s novelty and contributions.

Experimental design

The experimental design is sound and well-suited for addressing the research problem. The authors implemented the iBlast framework, applying it to MMseqs2 and Diamond, and benchmarked their performance against traditional methods using the Astral Scope dataset. The incremental approach is clearly explained, and the methodology allows for fair comparison between incremental and non-incremental methods.

The experimental setup (e.g., dividing the dataset into batches and using E-value thresholds) is appropriate for evaluating performance metrics such as hit count, processing time, and correlation. The authors also explored several correlation measures (Pearson, Kendall tau), which add robustness to their findings.

However, the methods section could benefit from further clarification on how the Spouge statistics are integrated into the incremental process.

Validity of the findings

The findings presented in the article appear to be valid, with evidence supported by both figures and tables. The incremental methods demonstrated substantial improvements in processing times and hit counts, as shown in Figure 1 and Table 1. The use of correlation metrics in Figure 2 shows that the incremental methods maintain consistency with their non-incremental counterparts.

The conclusion that incremental methods are more efficient while providing comparable accuracy is well-supported. The use of real datasets, coupled with the comprehensive evaluation of both time and accuracy metrics, strengthens the validity of the results.

Nevertheless, further validation across different datasets (besides the Scope Astral dataset) could provide more generalized support for the approach.

Additional comments

The article introduces a promising approach for protein sequence searching by integrating incremental learning with established methods such as MMseqs2 and Diamond. This approach addresses the challenges of large-scale genomic and proteomic data searches, and the results demonstrate clear advantages in efficiency and coverage.

One area for improvement is to include a more detailed discussion on the potential trade-offs between hit count and statistical significance, especially when E-values increase. This could help readers better understand the balance between comprehensive searches and potential false positives.

Reviewer 2 ·

Basic reporting

The references in the article contain grammar issues and inconsistencies in formatting.

Experimental design

• The log Discounted Cumulative Gain used to measure quality of the searches does not indicate biological relevance, as mentioned in the Discussion section

Validity of the findings

• There is no comparison between iSeqsSearch and other available methods mentioned in the article. The article mostly continues to compare incremental methods iMMseqs2, iDiamond, iBlastp vs non-incremental methods MMseqs2, Diamond, Blastp. There is little mention of the iSeqsSearch method in the figures.

• We would like to see how iSeqsSearch compares to other incremental methods. The iSeqsSearch method is not discussed in the Discussion or Results or Conclusions section. Please add in-depth discussion on method developed with results and comparison to the other incremental methods.

• There is minimal difference in time between MMSeqs2 and iMMSeqs2; Diamond and iDiamond shown in the Fig 5 in supplementary section.

Github:
• The github repo is missing Dockerfile to build the image. It is a standard practice to include the dockerfile.

• The requirements.txt file should include the biopython version mentioned in the article. Numpy package is mentioned in the article, but not provided in requirements file.

• Please provide instructions to create the m8e format introduced in the article. This is not mentioned in the github repository as well (in examples).

• Please mention python version clearly in the github repository.

• Please explain the optional parameters mentioned in the github repo.

• The examples mentioned in the github don’t work. There is no m8 file in source directory. I got an error:

in get_m8_record
with open(result_file, 'r') as file:
FileNotFoundError: [Errno 2] No such file or directory: 'source/blastp_astral_scope_result2.m8'

• Docker example does not work:

docker run --rm -v /home/user/data:/app/data comhyunwoo/incremental-protein-search:latest --default /app/data/blastp_astral_scope_result2.m8 /app/data/blastp_astral_scope_result3.m8 /app/data/merge_blastp_astral_scope_result23.m8 6205115 6211928
docker: Error response from daemon: Mounts denied:
The path /home/user/data is not shared from the host and is not known to Docker.
You can configure shared paths from Docker -> Preferences... -> Resources -> File Sharing.
See https://docs.docker.com/desktop/mac for more info.
ERRO[0000] error waiting for container: context canceled

Additional comments

There is a need to handle growing databases and the approach taken to develop new method is commendable. The use of Kendall tau and Pearson correlations to evaluate searches is helpful. The paper could discuss the applications of the method and suggest use cases. The iSeqsSearch method mentioned in the article needs to be compared to other published methods.

Annotated reviews are not available for download in order to protect the identity of reviewers who chose to remain anonymous.

---

## Round 0.2 · Minor Revisions

I agree with the reviewer. Since this is a bioinformatics tool for sequence analysis, the code has to be run by independent researchers. Please revise the manuscript by following and responding to the reviewer's comments and re-submit your manuscript after the revision.

Reviewer 2 ·

Basic reporting

Line 94: Does iSeqSearch use m8e or m8 or both? Please clarify as your next few lines explain the m8e format.

Experimental design

None to comment

Validity of the findings

- Please check the github repository as I am unable to run the code. I have attached detailed logs from my side. I have tested using both python 3.8 and python 3.9

- The dockerfile seems to be referencing a different repo (https://github.com/ece303/Incremental-Protein-Search.git) from the one mentioned in paper (https://github.com/EESI/Incremental-Protein-Search.git). Please consolidate code into one if possible (but this is very minor).

- The docker run does not work due to run_merge.sh error. Please see attached logs.
Here is the error:
File "source/main.py", line 67, in <module>
main()
File "source/main.py", line 60, in main
merge_m8e_files(file1, file2, output_file)
File "/Users/Incremental-Protein-Search/Incremental-Protein-Search/source/merger.py", line 87, in merge_m8e_files
rescale_evalues_m8(records1, part_db_len1, full_db_len)
File "/Users/Incremental-Protein-Search/Incremental-Protein-Search/source/merger.py", line 52, in rescale_evalues_m8
evalue = float(alignment[-2])
IndexError: list index out of range


- The readme.sh contains "comhyunwoo/incremental-protein-search" docker image from dockerhub repository which was last updated 7 months ago. I have tested both the image from dockerhub and building locally. I have attached the logs.

Annotated reviews are not available for download in order to protect the identity of reviewers who chose to remain anonymous.

---

## Round 0.3 · accepted · Accept

The manuscript has been well-revised and can be accepted now.